# Clinical and Socio-Demographic Variables Associated with the Diagnosis of Long COVID Syndrome in Youth: A Population-Based Study

**DOI:** 10.3390/ijerph19105993

**Published:** 2022-05-15

**Authors:** Eugene Merzon, Margaret Weiss, Beth Krone, Shira Cohen, Gili Ilani, Shlomo Vinker, Avivit Cohen-Golan, Ilan Green, Ariel Israel, Tzipporah Schneider, Shai Ashkenazi, Abraham Weizman, Iris Manor

**Affiliations:** 1Leumit Health Services, Tel-Aviv 6473817, Israel; emarzon@leumit.co.il (E.M.); svinker@leumit.co.il (S.V.); agolanchoen@leumit.co.il (A.C.-G.); igreen@leumit.co.il (I.G.); dr.ariel.israel@gmail.com (A.I.); 2Adelson School of Medicine, Ariel University, Ariel 4076414, Israel; shaias@ariel.ac.il; 3Cambridge Health Alliance, Cambridge, MA 02139, USA; margaret.weiss@icloud.com; 4Icahn School of Medicine at Mount Sinai, New York, NY 10029, USA; beth.krone@mssm.edu; 5ADHD Unit, Geha Mental Health Center, Petah Tikva 49100, Israel; weizmana@gmail.com (A.W.); dr.iris.manor@gmail.com (I.M.); 6Department of Psychiatry, Sackler Faculty of Medicine, Tel Aviv University, Tel Aviv 6997801, Israel; gili.2209@gmail.com; 7Department of Family Medicine, Sackler Faculty of Medicine, Tel Aviv University, Tel Aviv 69978, Israel; 8Clalit Health Services, Bnei Brak 5111501, Israel; tzipi.schneider@gmail.com

**Keywords:** long COVID, ADHD, COVID-19, morbidity

## Abstract

This study examines the demographic, clinical and socioeconomic factors associated with diagnosis of long COVID syndrome (LCS). Data of 20,601 COVID-19-positive children aged 5 to 18 years were collected between 2020 and 2021 in an Israeli database. Logistic regression analysis was used to evaluate the adjusted odds ratio for the characteristics of the COVID-19 infection and pre-COVID-19 morbidities. Children with LCS were significantly more likely to have been severely symptomatic, required hospitalization, and experienced recurrent acute infection within 180 days. In addition, children with LCS were significantly more likely to have had ADHD, chronic urticaria, and allergic rhinitis. Diagnosis of LCS is significantly associated with pre-COVID-19 ADHD diagnosis, suggesting clinicians treating ADHD children who become infected with COVID-19 remain vigilant for the possibility of LCS. Although the risk of severe COVID-19 infection and LCS in children is low, further research on possible morbidity related to LCS in children is needed.

## 1. Introduction

Phases of COVID-19 illness

Two years into the COVID-19 worldwide pandemic, the scientific community is still learning about intermediate and long-term morbidity from COVID-19. Many patients develop persistent symptoms following the acute phase of illness with the SARS-CoV-2 virus [1], with the acute phase defined as the first three weeks of infection [1]. The acute phase may present with mild symptoms [2] or entirely asymptomatic [3]. Symptoms completely resolve after the acute phase for fewer than half of patients [2,4]. Among some, a post-acute phase hyperinflammatory illness begins between weeks 2 to 4 post-infection [5]. Underlying characteristics of the immune system and prior exposure to Epstein Barr virus [6] and the viral RNA load in the bloodstream [7] are suggested to be predictive factors contributing to ongoing complications of COVID-19. Long COVID is a term used to describe all patients who continue to be symptomatic beyond the initial acute illness, either because the original infection did not resolve or because of post-infectious complications.

Many patients continue to experience symptoms through a ‘prolonged’ phase encompassed by the next 4 to 12 weeks [1]. These may continue unremitting symptoms or begin as a recurrence of symptoms after a brief latency period [1,5,8] even after a mild or asymptomatic first phase [2,3,4]. Some research indicates that this phase is more severe for individuals with mild initial phases of the disease [4]. Vaccination may improve many lingering somatic symptoms but no neurological symptoms among some patients [4]. Long haul COVID is a term used to describe all patients who continue to be symptomatic beyond the initial acute illness, either because the original infection did not resolve or because of post-infectious complications.

A third disease phase [9] has been identified, beginning 12 months post-infection [9]. Characterization of this phase is less well established, and the etiology is likely heterogeneous. Some proposed mechanisms of persistent pathology in this phase include ongoing sequelae of acutely damaged tissues, continued tissue damage by persistent infection, continued sequelae of immune dysregulation, and the onset of autoimmunity [10]. The various neurologic, psychiatric, pulmonary, cardiac, musculoskeletal, and gastrointestinal symptoms that persist in this post-acute phase may reflect these various etiologies or some predisposition factor of patient characteristics. Just as there are varying degrees of illness during the acute infectious period of COVID-19, so too there are varying degrees of illness in this chronic phase [11].

### 1.1. Defining the Syndrome

Currently, there is no universal nomenclature for the persistence of symptoms in the later phases of COVID-19 disease. The World Health Organization provided U09.9 as an ICD-10 code to facilitate diagnosis and billing for the care of symptoms, covering the terms long COVID, long haul, COVID-19 sequelae, post-acute sequelae of COVID-19 (PASC; [6]), post COVID-19 syndrome, and post-acute COVID-19 syndrome (PACS), among other terms. In the clinical and research reports discussing the phenomena, terms include persistent COVID-19 symptoms, post COVID-19 manifestations, long-term COVID-19 effects, and Late effects of COVID Infection (see the literature review by Yong, 2021 [10] for a review of proposed naming and descriptive conventions). For this article, we use the term “Long COVID Syndrome” (LCS) to describe the existence of signs and symptoms that develop following confirmed SARS-CoV-2 infection and continue for more than 12 weeks, and that is not explained by an alternative diagnosis [12]. Persistent physical symptoms consistent with LCS include fatigue, dyspnea, chest pain, cough, headache, and joint pain [13,14]. Common mental and cognitive symptoms include brain fog, insomnia, anxiety, and depression [9,15].

Some research indicates that the risk of developing this syndrome may be higher in hospitalized patients for more severe COVID-19 illness. Still, the syndrome is also prevalent in patients who have had mild or asymptomatic infection [2,4,16]. Some research suggests that females are more susceptible than males to LCS [2,17], although some research suggests that this represents a referral bias. It may be that increased LCS among women reflects increased participation in follow-up care rather than susceptibility to LCS [2]. Some research suggests that the syndrome is more prevalent among patients with preexisting chronic illnesses such as diabetes or cardiovascular disease [18]. A great deal of research indicates that the syndrome is more likely to occur with older age when contracting the infection, although there are reports of LCS among youth [2]. More work must be conducted to characterize and detect populations at risk of developing LCS for symptom monitoring and better prepare health service providers for the long-term needs of these patients and the large-scale impact resulting from population-wide exposures to the virus [19,20].

The heterogeneity of LCS is especially pertinent when studying a pediatric population who appear to have a mild initial illness compared with adults [21,22]. An important consideration in pediatric LCS is that illness occurs during critical developmental stages and processes. Studies are still conflicted about the risk of LCS in children and adolescents. Many youth experience symptoms after four weeks but less than 12 weeks post-acute infection [2,23]. Follow-up studies have found that 2.3% of the youth developed multiple inflammatory syndrome of COVID-19 (MIS-C). However, even after asymptomatic infection, both children with and without MIS-C experienced debilitating LCS up to nine months after the infection [24].

### 1.2. Neurodevelopmental Concerns

There have been recent reports of seizure activity among asymptomatic youth as young as 3 months old with the Omicron variant [25]. Among youth with severe infection, neurological sequelae are common [26]. Several studies suggest youth and adults may experience ongoing neurological sequelae of COVID-19, which occur more often among individuals with preexisting neurological conditions [26,27,28].

Attention deficit and hyperactivity disorder (ADHD) is the most common chronic neurodevelopmental disorder according to epidemiological research. Furthermore, literature from genetics (candidate gene research, GWAS, and pharmacogenomics) and brain imaging (PET, SPECT, MRI, task-based fMRI, and DTI) studies show it to be highly heritable and characterized by a variety of subtle vulnerabilities in the architecture of the brain, and in functional connectivity of neural networks [29,30]. As such, it is often associated with various cognitive and learning problems. 

Attention deficit and hyperactivity disorder (ADHD) is characterized by difficulty with attention, distractibility, organization, hyperactivity, and impulsivity [31]. Merikangas et al. (2010) studied the prevalence of different mental disorders among US children during 2001–2004. They found that the rate of ADHD in the general population was 15% in some populations of school-aged children, even with considerable under-diagnosis [32]. The somatic and mental comorbidities associated with ADHD are high [33,34,35]. Common somatic conditions associated with ADHD include asthma [36], atopic dermatitis [37], epilepsy [38], morbid obesity [34], and a predisposition to various infections [39]. Several studies report that ADHD is associated independently with increased rates of SARS-CoV-2 infection [27,28,40] and with a more severe course regarding symptomatology and the need for hospitalization, regardless of other background conditions [27,41]. This raises the question of whether ADHD might also associate with LCS. 

### 1.3. Objective 

The main objective of our study was to characterize the demographic, clinical and socioeconomic factors associated with diagnosis of LCS in the childhood population. The specific objective was to evaluate the possible association between diagnosis of LCS and previous diagnosis of ADHD.

We hypothesized that LCS might be associated with ADHD and that this association would be independent of other pre-COVID-19 comorbidities [42]. 

## 2. Methods 

Leumit Health Services (LHS) has a comprehensive online computerized database of patients’ demographics, medical visits, laboratory tests, hospitalizations, and medication prescriptions. Utilizing the LHS database, we conducted a population-based cross-sectional study.

The study population included all children aged 5–18 years (*n* = 20,601) who had a positive SARS-CoV-2 reverse transcriptase-polymerase chain reaction (RT-PCR) test between 1 February 2020 and 30 June 2021. The retrieved data from this population included: Demographic data: age, gender, and socioeconomic status (SES). SES was defined according to the child’s home address, using the Israeli Central Bureau of Statistics’ classification, including 20 subgroups. Classifications one to nine were considered low–medium SES, and ten to twenty were deemed medium–high SES;The characteristics of the COVID-19 infection: the severity of the COVID-19 infection (symptoms, hospitalization), the recurrences of the infection, and the existence of LCS. LCS was defined in LHS according to the WHO definition, and was coded “Late effects of coronavirus (COVID) infection” (139.2). Leumit prepared unique educational materials for primary care physicians based on the published studies’ clinical practice guidelines, editorials, and expert opinions for LCS management to increase LCS awareness;Pre-COVID-19 chronic comorbidities which may be related to COVID-19 or LCS. All the diagnoses were based on the International Classification of Disease, ninth revision (ICD-9) for somatic disorders, or tenth revision (ICD-10) codes for psychiatric disorders, as is required by all Israeli Health Maintenance Organizations (HMOs). The morbidities included chronic allergic rhinitis, atopic dermatitis, contact dermatitis, chronic urticaria, celiac, diabetes mellitus, bronchial asthma, depression, schizophrenia, and ADHD. Inflammatory bowel diseases (Crohn’s disease and ulcerative colitis) and psoriasis were not included due to very small numbers of affected children. In addition, body mass index (BMI) was not included due to missing data among the general children population during the study period.

ADHD was diagnosed according to the Israeli Ministry of Health criteria, following the international guidelines. The diagnosing physician was a senior physician specializing in ADHD (child or adult psychiatrists, child or adult neurologists, pediatricians, or family physicians with certified ADHD training). Diagnosis was based on longitudinal data, in multiple settings, using expert evaluation, according to the American Psychiatric Association’s Diagnostic and Statistical Manual criteria (DSM-IV or DSM-5). 

### 2.1. Ethics Statement 

The study protocol was approved by the Shamir Medical Center Review Board and the Research Committee of LHS.

### 2.2. Statistical Analysis

Continuous variables were analyzed using Student’s t-test or Mann–Whitney U test. The chi-square and Fisher’s Exact test for categorical variables were applied for binary and categorical variables. Stratified analyses were conducted and preliminary evaluation of risk estimates were made. Univariate and multivariate logistic regression model analyses were used to investigate associations between LCS and other demographic, clinical and socioeconomic factors. Odds ratio (OR) and 95% confidence intervals (CIs) are presented.

## 3. Results

The study population included *n* = 20,601 children aged 5–18 years infected by COVID-19, including *n* = 65 (0.31%) youth who were diagnosed with LCS. Approximately twenty percent (20.02%) of the overall sample of infected youth (*n* = 4125) were diagnosed with ADHD. 

*Age:* The mean age of children diagnosed with LCS were significantly older (M = 15.2 ± 2.51) than children without LCS diagnosis (M = 12.1 ± 3.45), t(20,6004) = −7.172, *p* < 0.000. The proportion of youth diagnosed with LCS was significantly older than the general sample of infected youth, X^2^ (1, *n* = 65) = 27.686, *p* < 0.000 was significant. In the age group 5–18 years, each year of age increased the association with LCS by 38%. 

There was no significant difference in gender representation among groups with LCS (53.81% male) as compared with the general population of infected youth (52.3% male), X^2^ (1, *n* = 65) = 0.062, *p* = 0.804. There was no significant difference between the SES of the LCS group (23% from a low SES family) and the general population (28% from a low SES family), X^2^ (1, *n* = 65) = 0.027, *p* < 0.870 (Table 1). 

The clinical characteristics of the study population and the characteristics of COVID-19 infection are shown in Table 2. The patients with LCS (10.77%) were significantly more likely to be symptomatic than were youth in the general sample (1.92%), X^2^ (1, *n* = 65) = 26.618, *p* < 0.001. Youth with LCS diagnosis more frequently required hospitalization (1.54%) as compared with the general population (0.03%), X^2^ (1, *n* = 65) = 43.594, *p* < 0.001). These youth with LCS more often had a recurrent acute infection within 180 days of original infection (1.54%) as compared with the general population (0.02%), X^2^ (1, *n* = 65) = 61.813, *p* < 0.001. 

Pre-existing comorbidities that were found to be more prevalent among patients with LCS were: schizophrenia X^2^ (1, *n* = 65) = 9.905, *p* < 0.002, chronic urticaria X^2^ (1, *n* = 65) = 9.096, *p* < 0.003, allergic rhinitis X^2^ (1, *n* = 65) = 9.960, *p* < 0.002, and ADHD X^2^ (1, *n* = 65) = 16.109, *p* < 0.000. *n* = 26 (0.13%) youth with schizophrenia were found in the general population, and *n* = 1 (1.54%) was diagnosed with LCS. *n* = 28 (0.14%) youth with chronic urticaria were found in the general population, and *n* = 1 (1.54%) was diagnosed with LCS. *n* = 571 (2.77%) youth with allergic rhinitis were found in the general population, and *n* = 6 (9.23%) were diagnosed with LCS. A statistically significant and clinically meaningful 40% of youth with LCS had ADHD diagnoses as compared with only 20% of youth in the general population. 

There were no significant differences among the groups related to pre-existing diagnoses of depression X^2^ (1, *n* = 65) = 1.344, *p* < 0.246, diabetes X^2^ (1, *n* = 65) = 1.759, *p* < 0.185, atopic dermatitis X^2^ (1, *n* = 65) = 0.626, *p* < 0.429, or celiac disease X^2^ (1, *n* = 65) = 0.317, *p* < 0.574. There was no significant difference (although there was a non-significant trend) among the groups in prevalence of pre-existing asthma X^2^ (1, *n* = 65) = 3.297, *p* < 0.069.

The crude and adjusted odds ratios between the demographic and the clinical variables and LCS are shown in Table 3. The demographic variables that were found to be associated with LCS were older age (adjusted OR = 1.38, 95% CI 1.24–1.54, *p* < 0.001), being symptomatic (adjusted OR= 5.29, 95% CI 2.36–11.84, *p* < 0.001), being hospitalized (adjusted OR = 44.70, 95% CI = 3.19–608.40, *p* < 0.001), and having recurrent acute infection(s) within 180 days (adjusted OR = 43.69, 95% CI = 3.71–514.24, *p* < 0.005). 

Three clinical syndromes were found to be associated with LCS: ADHD (adjusted OR = 2.02, 95% CI 1.19–3.40, *p* = 0.008, chronic allergic rhinitis (adjusted OR = 2.67, 95% CI 1.08–6.60, *p* = 0.033) and chronic urticaria (adjusted OR = 8.05, 95% CI 1.00–64.50, *p* = 0.049). Schizophrenia, depression, celiac disease, atopic dermatitis, and asthma were not found to have a statistically significant association with LCS. 

## 4. Discussion

Several variables were significantly associated with the likelihood of developing LCS. Three characteristics of acute COVID-19 infection predicted the risk of LCS: the severity of acute infection, being hospitalized, and recurrent acute infection(s). Only three pre-COVID-19 comorbidities were significantly associated with LCS: ADHD, chronic allergic rhinitis, and chronic urticaria.

Our study examined and controlled for obesity, diabetes, and other conditions known to be common risk factors for COVID-19 severity and hospitalization in our examination of the relationship between ADHD and LCS. The number of patients reporting some of these comorbidities was quite small. Thus, a significant association might have been missed. The current finding of the higher incidence of children with ADHD who are COVID-19 infected remained consistent with our and others’ previous observations of ADHD being a risk factor for infection [27,28].

### 4.1. ADHD

The finding that ADHD is significantly associated with LCS is consistent with previous findings that ADHD is associated with acquiring COVID-19 [27,28,40] and severity of COVID-19 [27,41]. ADHD is indeed associated with an increased rate of COVID-19 and greater severity of acute COVID-19, and this may partially account for the association with LCS. Alternatively, there may be a yet unidentified etiological relationship involving inflammatory processes or biological vulnerabilities. 

Several pathophysiological mechanisms are common to ADHD and LCS, which may be relevant to the associations between the two conditions. Notably, both manifest neuro-inflammatory processes. Part of the immune response to the SARS-CoV-2 is an increase in the inflammatory markers C-reactive protein (CRP), procalcitonin, and interleukin 6 (IL-6), which have been associated with gray matter tract changes, cerebrovascular flow, and microstructural changes to the frontal and limbic systems [43]. There are also several publications about ADHD and neuroinflammation [44]. A recent longitudinal study comparing LCS patients with the UK biobank control group demonstrated a significant reduction in grey matter thickness and tissue-contrast in the orbitofrontal cortex and parahippocampal gyrus; changes in changes markers of tissue damage in regions functionally connected to the primary olfactory cortex; and reduction in global brain size. These changes occurred even among people without a severe infection. It also found that the infected participants showed larger cognitive decline following infection [45]. Subjects with ADHD tend to present cortical thinning in the anterior cingulate cortex (ACC) and anterior insula, and increased diffusion tensor magnitude (a marker of tissue damage) in the ACC and the amygdala [46]. In both conditions brain changes present clinically as cognitive and executive dysfunctions, sometimes described as “brain fog”. 

The common pathophysiology of LCS and ADHD may explain why ADHD subjects are more vulnerable to LCS. LCS and ADHD share similar white matter abnormalities. A decreased density of axons has been found among LCS sufferers [47] as part of a demyelination process of formerly healthy neural tracts. Several researchers have found abnormal development of white matter structures in ADHD [29,30,48,49,50,51,52,53,54,55]. In addition, brain autoantibodies may be a part of the pathophysiology of LCS [56] and ADHD [57,58], leading to a shared vulnerability. 

### 4.2. Schizophrenia

In addition to ADHD, schizophrenia was also significantly associated with LCS. Schizophrenia’s inflammatory pathogenesis and neurovascular brain pathology may be relevant to this association [59]. Surprisingly, in contrast to the literature [60], depression was not found to be associated with LCS, which may be related to the small number of subjects with these disorders in the sample. 

### 4.3. Allergic Rhinitis

According to the literature, allergic diseases are associated with a higher risk of persistent symptoms at follow-up [61]. In our study, only chronic allergic rhinitis and chronic urticarial were of interest for their associations. The small sample size of other allergic diseases was too small, limiting this analysis’s power. 

In the British sample of *n* = 70,557, chronic allergic rhinitis lowered relative risk for testing positive for COVID-19 (relative risk [RR]: 0.75, 95% confidence interval [CI]: 0.69–0.81, *p* < 0.001) and long-term medication treatment with antihistamines, corticosteroids, and/or β2 adrenoceptor agonists did not impact risk [62]. Through much of the pandemic, there has been widespread difficulty in discriminating rhinitis exacerbations from early symptoms of infection [63]. This may have driven people with chronic allergic rhinitis to seek testing more often, resulting in over-testing of non-infected but symptomatic people with this condition, and acquisition of negative tests in that sample. Once diagnosed with COVID-19, though, many reports find that chronic allergic rhinitis is protective of more severe cases of COVID-19 [64]. One possible explanation for this being that chronic allergen exposure in allergic rhinitis is associated with decreases in ACE2 receptors in the upper respiratory system, which deprives COVID-19 of entry points [64].

### 4.4. Chronic Urticaria

Chronic urticaria is common and of heterogeneous etiology. To date, there have been two studies examining treatment of chronic urticarial as a determinant of COVID-19 infection severity, and both found that urticarial treatment did not impact the course of COVID-19. A study of *n* = 233 patients with chronic urticaria found no increase in susceptibility for, or severity of, COVID-19 infection between patients treated for chronic urticaria with IgE antibodies (omalizumab) or traditional H1 antihistamines [65]. In a pre-print report examining COVID-19 among *n* = 370 patients with chronic urticaria, there were no gender, treatment, or serum IgE influences on course of COVID-19; however, higher eosinophil count was significantly associated with COVID-19 infection [66]. This may indicate a role for eosinophilic pathologies in COVID-19 risk within this heterogeneous disorder. 

### 4.5. Limitations

Our study has several limitations. First, despite the relatively small sample size, the association between LCS and ADHD was robust enough to be detected. 

We are still in the early stages of clinical awareness and experience with LCS, so we have a high likelihood of failing to identify many cases. In the same way, we are in the early stages of research on LCS in general, and in children in particular. This means we lack strong consensus definitions for acute, post-acute, chronic, and recurrent COVID-19 symptoms.

## 5. Conclusions

There are several important implications to our study. First, our findings indicate the need to raise clinical awareness that LCS is real and can be present in children. This also impacts public health policy. Although the risk of severe illness and hospitalization is low in children, if COVID-19 or Omicron can precipitate chronic illness, this could become a source of significant long-term morbidity. Lastly, the association between LCS, ADHD, and urticaria is of heuristic interest in suggesting possible common pathophysiology. The presentation of ‘brain fog’ as a common complaint in LCS indicates that the underlying neurological substrate of cognitive dysfunction may be similar in both ADHD and LCS. Thus, awareness of the association of ADHD and LCS may facilitate early detection of LCS in the ADHD population and intervention accordingly.

## Figures and Tables

**Table 1 ijerph-19-05993-t001:** Demographic characteristics of the study sample (*n* = 20,601).

Demographics	Total Sample with COVID-19	Sample with Long COVID	Sig.
*n* (%)	20,601 (99.69%)	65 (0.31%)	
Age (years) Mean ± SD	12.1 ± 3.45	15.2 ± 2.51	*p* < 0.000
**Age Category *n* (%)**			*p* = 0.000
<12 years	8886 (43.13%)	7 (10.77%)	
12–18 years	11,715 (56.87%)	58 (89.23%)	
**Gender *n* (%)**			*p* = 0.804
Male	10,775 (52.30%)	35 (53.81%)	
Female	9826 (47.70%)	30 (46.15%)	
**SES *n* (%)**			*p* = 0.870
Low SES	14,826 (71.97%)	50 (76.92%)	
High SES	5775 (28.03%)	15 (23.08%)	

Note. SES = socioeconomic status. Significance is reported for *t*-tests and chi-square.

**Table 2 ijerph-19-05993-t002:** Clinical characteristics of the study sample (*n* = 20,601), analyzed by Chi^2^.

Variable	Total Sample with COVID-19 *n* (%)	Sample with Long COVID *n* (%)	Sig.
Symptomatic COVID-19 infection (%)	395 (1.92%)	7 (10.77%)	0.000
Hospitalization due to COVID-19	6 (0.03%)	1 (1.54%)	0.000
Recurrent infection within 180 days	4 (0.02%)	1 (1.54%)	0.000
ADHD	4125 (20.02%)	26 (40.00%)	0.000
Depression	105 (0.51%)	1 (1.54%)	0.246
Chronic allergic rhinitis	571 (2.77%)	6 (9.23%)	0.002
Chronic urticaria	28 (0.14%)	1 (1.54%)	0.003
Celiac	182 (0.88%)	1 (1.54%)	0.574
Schizophrenia	26 (0.13%)	1 (1.54%)	0.002
Type 1 Diabetes Mellitus	91 (0.44%)	1 (1.54%)	0.185
Asthma	2835 (13.76%)	14 (21.54%)	0.069
Atopic dermatitis	678 (3.29%)	1 (1.54%)	0.429

ADHD = attention deficit and hyperactivity disorder.

**Table 3 ijerph-19-05993-t003:** Logistic regression for crude and adjusted odds ratios for patients with long COVID syndrome, demographic and clinical characteristics.

	Crude			Adjusted		
Variable	OR	95% CI (LL, UL)	*p*	OR	95% CI (LL, UL)	*p*
Symptomatic COVID-19 infection	6.17	(2.80, 13.61)	0.00	5.29	(2.36, 11.84)	0.000
Hospitalization due to COVID-19 infection	53.63	(6.36, 451.84)	0.00	44.70	(3.19, 625.40)	0.005
Recurrent infection within 180 days	80.45	(8.87, 729.76)	0.00	43.69	(3.71, 514.24)	0.003
Age	1.42	(1.27, 1.58)	0.00	1.38	(1.24, 1.54)	0.000
Male gender	1.06	(0.65, 1.73)	0.80	0.87	(0.52, 1.45)	0.616
Low–medium SES	1.05	(0.56, 1.98)	0.87	1.17	(0.61, 2.23)	0.633
ADHD	2.66	(1.61, 4.37)	0.00	2.02	(1.19, 3.40)	0.008
Chronic urticaria	11.48	(1.53, 85.65)	0.017	8.05	(1.00, 64.50)	0.049
Chronic allergic rhinitis	3.56	(1.53, 8.29)	0.00	2.67	(1.08, 6.60)	0.033
Schizophrenia	12.36	(1.65, 92.48)	0.014	3.16	(0.30, 32.63)	0.333
Depression				1.57	(0.21, 11.84)	0.657
Asthma	1.72	(0.95, 3.11)	0.07	1.44	(0.77, 2.70)	0.247
Celiac	1.75	(0.24, 12.70)	0.57	1.47	(0.19, 10.83)	0.378
Atopic dermatitis	0.45	(0.06, 3.31)	0.440	0.40	(0.05, 2.99)	0.378

Note. CI = confidence interval; LL = lower limit; UL = upper limit; ADHD = attention deficit hyperactivity disorder; SES = socioeconomic status.

## Data Availability

Because of ethical and privacy issues, data cannot be shared.

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
