# Peer review of "Clinical and Socio-Demographic Variables Associated with the Diagnosis of Long COVID Syndrome in Youth: A Population-Based Study"

_ijerph, 2022, doi:10.3390/ijerph19105993_

Round 1

Reviewer 1 Report

The article documents the foreseeable association, based on the probable common pathophysiology, of suffering from LCS in pediatric patients previously diagnosed with ADHD. It is properly documented and well written.

I have not located the reference: Gupta, N. et al. Clinical Guidelines: Long COVID. 2021. This reference should be corrected or deleted.

No other references are available (e.g. Pubmed) to ADHD – SARS-COV-2, nor to ADHD - schizophrenia.

Author Response

Reviewer 1 commented:

  • I have not located the reference: Gupta, N. et al. Clinical Guidelines: Long COVID. 2021. This reference should be corrected or deleted.

RESPONSE: This reference is not indexed in PubMed, but lives as a clinical guideline on the OHSU website. We are following the reviewer’s recommendation to remove this reference (paragraph 2, page 4, ref #44).

  • No other references are available (e.g. Pubmed) to ADHD – SARS-COV-2, nor to ADHD - schizophrenia.

RESPONSE: We initially cited a preprint of the JAMA article (reference #46, page 8, paragraph 5 in the original text) that linked inflammatory processes with neuropsychiatric disorders. That preprint is now published and indexed on PubMed, and we have corrected the citation, as follows:

Williams, J. A., Burgess, S., Suckling, J., Lalousis, P. A., Batool, F., Griffiths, S. L., Palmer, E., Karwath, A., Barsky, A., Gkoutos, G. V., Wood, S., Barnes, N. M., David, A. S., Donohoe, G., Neill, J. C., Deakin, B., Khandaker, G. M., Upthegrove, R., & PIMS Collaboration (2022). Inflammation and Brain Structure in Schizophrenia and Other Neuropsychiatric Disorders: A Mendelian Randomization Study. JAMA psychiatry, e220407. Advance online publication. https://doi.org/10.1001/jamapsychiatry.2022.0407

Reviewer 2 Report

The work presented for review is another publication on this topic by an experienced team of researchers. The manuscript's strengths are:
- an innovative and little-researched topic
- very well collected data from the LHS database
- an excellent methodological workshop.
Although a controversial hypothesis (it seems impossible to make the severity of the COVID course of SARS-CoV-2 infection independent of the previous history of the disease, especially concerning inflammatory diseases), the authors have the right to it.
Descriptions and statistical analyzes require verification. Please check the compliance of the descriptions in the text with the data contained in tables 1 and 2.
The authors declare a study of 20.666 children (65 LCS) 20.601 no LCS) of which 17% with ADHD (out of the number of 20666, 3513 children, not 4125. In Table 2, the sum of children with ADHD LCS (n = 26) + no LCS (n = 4125) gives the total number 4151. Such discrepancies suggest a mistake at the initial stage of group qualification and undermine the correctness of further calculations.
The leading topic of the article is the relationship between ADHD and LCS. While in the calculations presented in table no 3, the ADHD diagnosis was treated as one of the variables. It should be focused on comparing ADHD with LCS vs. ADHD with no LCS and comparing it to children without ADHD.
The awareness of a relatively small group of crucial patients determines the pilot nature of the study, which should be reflected in the title.

Author Response

  • Although a controversial hypothesis (it seems impossible to make the severity of the COVID course of SARS-CoV-2 infection independent of the previous history of the disease, especially concerning inflammatory diseases), the authors have the right to it.

RESPONSE: Thank you for your comment regarding our hypothesis. This hypothesis is based on emerging, published data from several large scale epidemiological studies that we cite in the manuscript. These studies use electronic health records and one used MRI + prospectively collected behavioral data (in the British cohort study) to show that brain changes and symptoms of illness associated with LCS are not clearly dependent on prior health history or severity of SARS-COV-2 infection. Please see references 28, 29, 41, 42, 48, 49, 50, 51, 52, and 61, in particular.

  • Descriptions and statistical analyzes require verification. Please check the compliance of the descriptions in the text with the data contained in tables 1 and 2.

RESPONSE: We revised the tables 1 and 2, and their descriptions in the text (Results section, pages 4 through 7 for edits) to the best practice standard of APA 7 style.

  • The authors declare a study of 20.666 children (65 LCS) 20.601 no LCS) of which 17% with ADHD (out of the number of 20666, 3513 children, not 4125. In Table 2, the sum of children with ADHD LCS (n = 26) + no LCS (n = 4125) gives the total number 4151. Such discrepancies suggest a mistake at the initial stage of group qualification and undermine the correctness of further calculations.

RESPONSE: Thank you for pointing out the error. We have corrected the manuscript to reflect the accurate sample sizes (Total N=20,601, n=4125 (~20%) in the LCS group.) All analyses were conducted using the correct sample sizes, and we have corrected the reporting (tables and in text in the Results section, pages 4 through 7).

  • The leading topic of the article is the relationship between ADHD and LCS. While in the calculations presented in table no 3, the ADHD diagnosis was treated as one of the variables. It should be focused on comparing ADHD with LCS vs. ADHD with no LCS and comparing it to children without ADHD.

RESPONSE: In this manuscript, we checked the factors associated with LCS (demographics, features of COVID, comorbid/chronic diseases, etc.) and compared the proportion of ADHD among children who had and did not have an LCS diagnosis.

  • The awareness of a relatively small group of crucial patients determines the pilot nature of the study, which should be reflected in the title.

RESPONSE: This study is a records review that uses well-established epidemiological research methodology to describe the clinically observed phenomenon. We, of course, would like to do additional research among other populations, or further prospective research using additional methodologies. This study, however, is not a feasibility study and by definition, is not a pilot study.

Reviewer 3 Report

Thank you for the opportunity to review this paper.

It is a well written and honest presentation of the current limited knowledge of long term COVID. there are some spacing issues with one paragraph and table. Please correct these. there are minor errors in sentences/word order. pg 4 ...who were had COVID needs correcting; pg 8 "...of acute COVID, this may result in part account for the association with LCS. Alternatively, there it is also possible..."; ref 5,7,9,16,20,27,28,31,40,42,43,53,61 all have title capitalizations that should be lower case.

Otherwise very well written.

Author Response

  • There are some spacing issues with one paragraph and table. Please correct these.

RESPONSE: Thank you for pointing this out. We have made revisions. To both tables 1 and 2, and to several paragraphs that were (or became as we edited) affected by content control features of Microsoft Word.

  • There are minor errors in sentences/word order. pg 4 ...who were had COVID needs correcting; pg 8 "...of acute COVID, this may result in part account for the association with LCS. Alternatively, there it is also possible..."

RESPONSE: Thank you for pointing these out. We have made revisions to correct minor errors in word order on page 4 Methods and Statistical analyses and page 8 Discussion section as you point out, and also to page 3 section 1.2 paragraph 2.

  • ref 5,7,9,16,20,27,28,31,40,42,43,53,61 all have title capitalizations that should be lower case.

            RESPONSE: Thank you for pointing these out. We have made revisions to the reference list.

Round 2

Reviewer 2 Report

I accept the authors' corrections and clarifications with one exception. I affirm the need to complete the paragraph:

  • The leading topic of the article is the relationship between ADHD and LCS. While in the calculations presented in table no 3, the ADHD diagnosis was treated as one of the variables. It should be focused on comparing ADHD with LCS vs. ADHD with no LCS and comparing it to children without ADHD.

Data may be included as a supplement.

Author Response

We thank you and Reviewer #2 for your comments toward improving our manuscript. We have taken the following steps to resolve the reviewer’s concerns:

Specific concerns were cited:

Reviewer 2 commented: I accept the authors' corrections and clarifications with one exception. I affirm the need to complete the paragraph:

The leading topic of the article is the relationship between ADHD and LCS. While in the calculations presented in table no 3, the ADHD diagnosis was treated as one of the variables. It should be focused on comparing ADHD with LCS vs. ADHD with no LCS and comparing it to children without ADHD.

RESPONSE:

The main objective of our study was to characterize the demographic, clinical and socioeconomic factors associated with diagnosis of LCS in childhood population. The specific objective was to evaluate the possible association between diagnosis of LCS and previous diagnosis of ADHD. We hypothesized that LCS might be associated with pre-COVID ADHD diagnosis and that this association will be independent of others pre-COVID comorbidities. Utilizing LHS database we conducted a population-based cross-sectional study, while the study population included all children aged 5-18 years (n=20,601) who were had a positive SARS CoV-2 during the study period. LCS was defined as a dependent variable and we conducted univariate and multivariate logistic regression models to investigate an independent associations between LCS and other demographic, clinical and socioeconomic factors. Although ADHD was a specific objective of our research, our design found that other clinical syndromes were found to have even stronger association with LCS chronic urticaria (adjusted OR= 8.05, 95% CI 1.00- 64.50, p=.049) and chronic allergic rhinitis (adjusted OR=2.67, 95% CI 1.08-6.60, p=.033). We made changes in Title, Abstract, Methods, and discussion sections of the manuscript to clarify.